# Boost Your Medical Deep-Learning Training By Lazy Loading

**Chenglong Wang**[1]                                                     CLWANG@PHY.ECNU.EDU.CN
[1] *Shanghai Key Laboratory of Magnetic Resonance, East China Normal University, China*

**Chengxiu Zhang**[1]                                                     CXZHANG@PHY.ECNU.EDU.CN
**Yun Liu**[2]                                                                      YUNL@NVIDIA.COM
[2] *NVIDIA Corp.*

**Guang Yang**[1]                                                          GYANG@PHY.ECNU.EDU.CN

**Editors:** Under Review for MIDL 2024

## Abstract

In recent years, the growing volume size of medical datasets has posed a significant challenge for deep learning training pipelines, often leading to inefficiencies stemming from data I/O bottlenecks. Addressing this issue, we present a simply yet effective trick, *lazy loading* strategy, leveraging memory-mapping mechanisms to boost training processes. By dynamically loading only the target slices of large medical datasets into active memory, our method minimizes the reading time and conserves memory. This paper mainly aims to remind community to realize the advantages of the lazy loading strategy, which could substantially boost the efficiency of deep learning training process in the medical domain.

**Keywords:** Deep-Learning Efficiency, Memory-Mapping, I/O Optimization

## 1. Introduction

With the demonstrable success of large foundation models, it is clear that leveraging extensive training datasets can produce superior performance improvements. This trend towards grand-scale data not only holds promise but also introduces notable challenges, particularly for the training of medical imaging models. Medical datasets are generally large due to the high-resolution nature of medical images, requiring a significant amount of computational resources for efficient data handling and processing.

To address these challenges, innovations have been made to enhance medical image data input/output (I/O) operations. Frameworks such as MONAI (Cardoso et al., 2022) have developed specialized datasets - CacheDataset, PersistentDataset, and SmartCacheDataset among others - that aim to optimize data loading efficiency and resource utilization. However, these approach still face limitations when dealing with exceptionally large datasets when computational resources are constrained. Other approach such as "GPUDirect Storage" (GDS) technique aims to accelerate the data loading through a direct memory access from GPU, avoiding a bounce buffer through the CPU. Nevertheless, the efficiency of GDS is still compromised when training with substantial volumes of data.

Due to the typically large size of medical images, random cropping strategy is frequently adopted. Traditional eager loading strategies, wherein entire data is loaded into memory, can lead to substantial system strain and efficiency gap. For many patch-based training tasks, having the entire dataset in memory tends to be unnecessary. To mitigate these

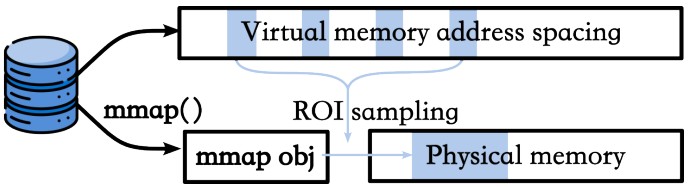

Figure 1: Simple illustration of lazy loading using memory-mapping technique.

limitations, this work presents a lazy loading approach that utilizes memory-mapping to optimize data loading during training. In the context of medical deep learning, where datasets comprise high-resolution images requiring considerable memory demand. Memory-mapping permits sections of a dataset to be selectively read into memory, improving not only memory utilization but also training speed. *Lazy loading* could be a fundamental approach in enhancing the efficiency of deep learning training routines in medical imaging, ensuring an optimization in performance without compromising the quality and integrity of the learning process.

## 2. Method

To address the challenge of handling sizable medical datasets during deep learning training, our method employs a tailored lazy loading strategy using memory-mapping techniques. Memory-mapping technique has been widely used in many field, its application in accelerating the loading of medical images is a highly suitable use case. Simple workflow of lazy-loading strategy is illustrated in Figure 1.

In the lazy loading strategy, instead of loading the entire image into memory, we directly map the image file into the memory, enabling fast access to image data stored in local storage. Detailed steps are described in Algorithm 2. For each image, memory-mapped object is firstly created. Then, necessary metadata is extracted from memory-mapped object without reading the complete file, and calculate ROIs based on the meta information with given patch size. We then retrieve the actual data corresponding to each ROI directly from local storage.

Memory-mapping technique not only reduces memory constraints but also decrease I/O operation time, thereby accelerating the training pipeline. The complexity of lazy-loading is in managing memory-mapping to function as an optimized data retrieval process, specifically designed for 3D medical images. Fortunately, third-party libraries like NumPy (Harris et al., 2020) and NiBabel (Brett et al., 2024) offer various memory-map APIs, which effectively aid in implementing this strategy. The pre-release version is publicly avaiable on Github[1].

## 3. Results

We evaluated the efficiency of lazy-loading for spleen segmentation from 3D CT scans (MSD challenge), and lung nodule detection from the LIDC public dataset. For spleen segmentation, we trained on 32 CT images with randomly cropped patches of $96 \times 96 \times 96$

---

1. https://github.com/Project-Strix/MONAI

---

**Algorithm 1** Lazy Loading (Random)

---

**Input: D**, medical image paths; $s$ patch size
**Output: P**, output patches
**Function** `Main`**:**

   **P** ← ∅
   **foreach** $d$ *in* **D do**

      $mmapped\_obj$ ← `mmap`$(d)$ ;       `// Get memory-map object to address space`
      $meta$ ← `GetMetaInfo`$(mmapped\_obj)$ ;         `// Get meta info of data`
      $ROIs$ ← `RandomSample`$(meta, s)$ ;         `// Random ROIs generation`
      **foreach** $ROI$ *in* $ROIs$ **do**

         $p$ ← `mmap`$(mmapped\_obj, ROI)$;   `// Retrieve ROI data from mmapped obj`
         **P** ← **P** ∪ {$p$}
      **end**

   **end**
   **return P**

---

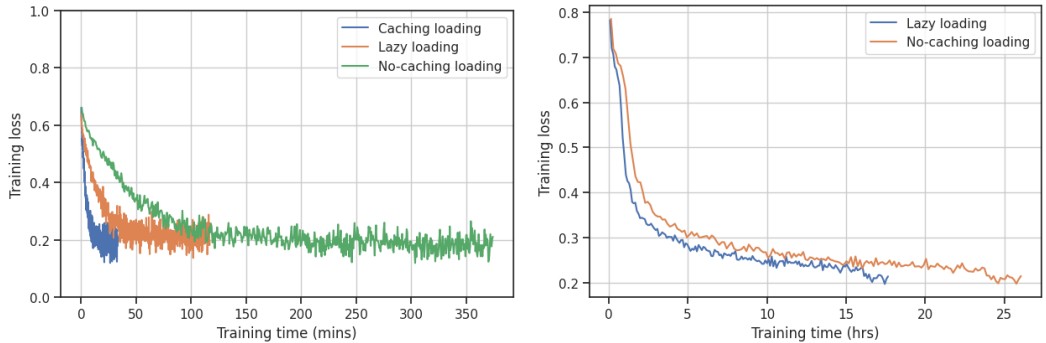

Figure 2: Training time of spleen segmentation (left) and lung nodule detection (right).

for 600 epochs on a single NVIDIA RTX 3090 GPU, comparing three loading strategies: no-caching loading, fully cached loading, and lazy-loading. Lung nodule detection training employed 830 CT images with $128 \times 128 \times 64$ crops for 300 epochs on a single NVIDIA A100 GPU, where we only compared the baseline with lazy-loading since the entire dataset is too large too be fully cached. The results are shown in Figure 2. Comparing to the baseline no-caching loading, our lazy-loading achieved 3.2x and 1.5x speedup of segmentation and detection task, respectively. It should be noted that acceleration rates may vary according to the specific tasks and configurations involved..

## 4. Conclusion

Lazy-loading strategy could be a better fundamental loading strategy in patch-based medical deep-learning training routines than the traditional no-caching loading. However, when sufficient physical memory is available, full caching loading remains the optimal choice. The integration of the lazy-loading strategy with the GDS technique is one of future works that could further optimize training efficiency.

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
