# OpenReview forum: "Boost Your Medical Deep-Learning Training By Lazy Loading"
_MIDL.io/2024/Short_Papers — MIDL 2024 Short Papers_

### Official Review · Reviewer_2dqv · 2024-04-16

**Confidence:** 5
**Final Rating:** 4

**Review:**

This paper explores different data loading strategies for medical deep-learning training routines, specifically comparing the lazy-loading strategy to traditional no-caching and full caching methods. It identifies the contexts in which each strategy excels, noting that lazy-loading can be superior in resource-constrained environments, whereas full caching remains optimal when sufficient memory is available. Furthermore, the paper suggests a potential advancement in the form of integrating lazy-loading with the GDS technique to enhance training efficiency.

The limitation of this work includes the lack of comparison and discussion with the distributed database and fiber-connected distributed I/O system for training today's large-scale AI foundation models.

---

### Decision · Program_Chairs · 2024-04-26

Accept